# Effects of Acute Low-Frequency Pulsed Electromagnetic Field Therapy on Aerobic Performance during a Preseason Training Camp: A Pilot Study

**DOI:** 10.3390/ijerph18147691

**Published:** 2021-07-20

**Authors:** Nauris Tamulevicius, Tanuj Wadhi, Guillermo R. Oviedo, Ashmeet S. Anand, Jung-Jung Tien, Fraser Houston, Eric Vlahov

**Affiliations:** 1Department of Health Sciences and Human Performance, The University of Tampa, Tampa, FL 33606, USA; tanuj.wadhi@gmail.com (T.W.); ashmeetanand1@gmail.com (A.S.A.); fhouston@ut.edu (F.H.); evlahov@ut.edu (E.V.); 2Faculty of Psychology Education and Sport Science Blanquerna, University Ramon Llull, 08022 Barcelona, Spain; guillermorubeno@blanquerna.url.edu; 3Department of Internal Medicine, University of Central Florida/HCA GME Consortium, Greater Orlando, FL 32827, USA; Jung-Jung.Tien@ucf.edu

**Keywords:** aerobic performance, low-frequency pulsed electromagnetic field therapy, ventilatory threshold, runners

## Abstract

Bio-electromagnetic-energy-regulation (BEMER) therapy is a technology using a low-frequency pulsed electromagnetic field (PEMF) in a biorhythmic format. BEMER has been shown to optimize recovery and decrease fatigue by increasing blood flow in microvessels. Our aim was to determine its effects during preseason training in endurance athletes. A total of 14 male cross-country runners (19.07 ± 0.92 y.o.) were placed in either the intervention (PEMF; *n* = 8) or control (CON; *n* = 6) group using a covariate-based, constrained randomization. Participants completed six running sessions at altitudes ranging from 881.83 (±135.98 m) to 1027.0 (±223.44 m) above sea level. PEMF group used BEMER therapy before and after each training session, totaling 12 times. There were no significant changes in absolute or relative VO2Peak, ventilation or maximum respiration rate for either the PEMF or CON group (*p* > 0.05). There was a significant effect of time for absolute and relative ventilatory threshold (VT), and maximum heart rate, heart rate at VT and respiration rate at VT. This study was the first of its kind to study PEMF technology in combination with elevated preseason training. Results indicate some evidence for the use of PEMF therapy during short-term training camps to improve VT.

## 1. Introduction

Pulsed electromagnetic field (PEMF) treatment has been used for therapeutic purposes for almost half a century [1]. The application of external electrical, mechanical, and/or electromagnetic energy to the area of injury induces changes to the cell environment and restores the integrity and function of tissues within the organisms. This form of therapy has also been approved for the treatment of delayed and nonunion fractures in humans by the United States Food and Drug Administration since 1979 [2]. PEMF was also found to be effective for (1) pain management and edema after soft-tissue injury, (2) osteoarthritis-related injuries, (3) repairing ligaments and tendons, (4) wound [3,4,5,6] and bone fracture healing [7,8,9], (5) reducing subjective soreness [10,11], and (6) promotion of regeneration of nerves [4,12,13,14,15].

However, these devices are primarily advertised and distributed over the internet and are often used without medical supervision. According to their manufacturers, the therapeutic indications cover a wide range of diagnoses such as insomnia, back pain, osteoporosis, arthritis, cardiovascular disorders, and neurodegenerative diseases. In addition, whole-body PEMF mats are also frequently proposed as wellness items [16]. Thus far, a biological mechanism that could explain the therapeutic effects of whole-body PEMF devices has not been proven; yet, the manufacturers postulate a large variety of mechanisms including the stimulation of cell protein synthesis, antioxidants, and osteoblasts, as well as increases in microcirculation, leading to improved oxygen supply and enhanced immunological functions [17]. Bio-electromagnetic-energy-regulation (BEMER) therapy is a technology using low-frequency PEMF of flux density 35–50 µTesla in a biorhythmic format [18] and has been shown to optimize recovery and decrease fatigue by increasing blood flow, specifically in microvessels, which make up a majority of the vasculature [19].

Interestingly, the positive effects seen from PEMF therapy have primarily been seen in extreme situations, while effects for healthy adults have been inconclusive [17]. Athletes often undergo training camps in the preseason phase, to perform intensified training loads, hoping to maximize adaptations for the upcoming competitive season [20]. The onset of such a training regimen, following a period of sedentary behavior, represents a significant physical challenge to athletes. Previous evidence has shown this type of training camp produces a state of physiological fatigue, reduces subjective wellness of athletes [20], and can also affect sleep quantity and quality [21], further affecting recovery.

Methods that permit an expedited physiological adaption to a training load could be advantageous to athletic performance. Considering the previous evidence, along with the common-place usage of these devices in practical settings, PEMF, specifically BEMER-PEMF therapy, could potentially improve athletic performance when the athletes return to the regular training season by increasing recovery and decreasing fatigue during the training camp. Previous research has shown the positive effects of PEMF on recovery from exercise. For example, Grote et al. [22] showed improved autonomic recovery during short-term usage of PEMF after physical exercise, while Rasmussen et al. [11] and Jeon et al. [10] showed decreased symptoms of DOMS after usage of PEMF.

Performance in aerobic sports such as cross-country running is typically measured through multiple cardiopulmonary parameters [23,24,25]. A key determinate of endurance performance is the measurement of maximum aerobic capacity (VO_2Max_) [26], and improvements for this parameter are dependent on factors including training intensity over multiple sessions [27]. Improved recovery times could therefore potentially increase VO_2Max_ at a quicker rate, providing a competitive edge to the athlete. Another predictor of endurance performance, especially in long-distance runners is the ventilatory threshold (VT), defined as the intensity at which ventilation increases disproportionately to oxygen consumption, which is also expressed in relation to VO_2Max_ percentage [28].

Interestingly, despite the common use of acute PEMF therapy in practical settings, no study, to the best of the author’s knowledge, has investigated the effects of this technology on sports performance. For this reason, the current pilot study is intended to explore the viability of PEMF for athletic performance. Therefore, the aim of the current pilot study was to see the effects of BEMER-PEMF therapy on aerobic performance during a collegiate preseason training camp in endurance athletes. Given the above evidence of PEMF-induced recovery and regeneration, we hypothesized that BEMER-PEMF would result in improved aerobic performance parameters.

## 2. Materials and Methods

### 2.1. Study Design

The pilot study utilized a randomized-controlled study design to measure the effects of the BEMER therapy. Participants were divided into either the intervention group, who received the BEMER therapy, or a control group, who did not receive the intervention during a preseason elevated training camp. Training intervention, sleep duration, nutrition, and other environmental factors were similar for all the participants throughout the duration of the study.

### 2.2. Participants

A total of 14 male National Collegiate Athletic Association (NCAA) Division 2 cross-country runners (age: 19.07 ± 0.92 y.o.) from the university’s cross-country running team, with initial peak aerobic capacity (VO_2Peak_) of 73.13 ± 5.65 mL/kg/min, participated in the study. Participants were placed in either the intervention (PEMF) or control (CON) group using a covariate-based, constrained randomization, executed via a computer program [29]. The VO_2Peak_ of the participants from baseline testing (i.e., the covariate) were inserted into the program, which then looped through randomly generated groups. The final groups’ output by the algorithm was the combination in which the two groups met the predefined criterion (i.e., <1% coefficient of variation between groups). If the criterion could not be met, the output was the combination of groups with the lowest coefficient of variation. All participants signed written informed consents prior to participation, and the study was approved by the university’s institutional review board.

### 2.3. Assessments

Participants were tested at sea level in the university’s Human Performance Lab for their peak aerobic capacity using a metabolic system (ParvoMedics TrueOne 2400, Salt Lake City, UT, USA), which was calibrated prior to each test. Data were also collected for absolute VO_2Peak_ (AbsVO_2Peak_ [L/min]), relative VO_2Peak_ (RelVO_2Peak_ [mL/kg/min]), ventilation (VE [L/min]), absolute ventilatory threshold (VT_Abs_ [L/min]), relative VT (VT_Rel_ [% of VO_2Peak_]), heart rate at VO_2Peak_ (HR_Max_ [Beats/min]), heart rate at VT (HR_VT_ [Beats/min]), respiration rate at at VO_2Peak_ (RR_Max_ [Breaths/min]), and respiration rate at VT (RR_VT_ [Breaths/min]).

The testing protocol consisted of an incremental graded exercise test (GXT) performed on a treadmill (TuffTread, Conroe, TX, USA). GXT started with a warm-up at 4.8 km/h for 3 min, followed by gradual increments in speed over the next 5 min (i.e., until minute 8) to achieve a speed that the participant(s) could maintain for the remainder of the test (~70% of heart rate reserve [HRR]). After 8 min, the speed was held constant, while the inclination grade was increased by 2% every 2 min. The test was terminated when the participant(s) reached volitional failure or if a respiratory exchange ratio (RER) of ≥1.15 was achieved.

The ventilatory threshold was determined using a data plot created by the metabolic system software [ParvoMedics, OUSW4.3.4] by two independent researchers and confirmed by a third researcher who was not involved in the testing process. VT was considered as the point when VCO_2_ started increasing disproportionately to VO_2_ (Wasserman). The final VT values used for analysis were those concurred by all three researchers.

### 2.4. Intervention

Runners traveled from sea level to an altitude of 1322 m above sea level for 6 days of training. They completed 6 training sessions of running at altitudes ranging from 881.83 ± 135.98 m to 1027.0 ± 223.44 m above sea level. The team trained together, and both groups had the same training per day. On average, each training session was 64.50 ± 19.05 min at a speed of 3.62 ± 0.44 m/s (13.02 ± 1.60 km/h), covering an average distance of 16,415 ± 2950 m. Altitude was measured using two separate GPS altitude devices (Ambit 3 Sport, Suunto Oy, Vantaa, Finland). The daily training regime for the team is presented in Table 1. The athletes trained on their own and workload was not monitored prior to the pre-season camp in compliance with NCAA regulations.

Eight men (age: 19.01 ± 0.69 years; VO_2Peak_: 72.90 ± 6.39 mL/kg/min) were assigned to the PEMF group, while six (age: 19.12 ± 0.99 y.o.; VO_2Peak_: 72.41 ± 4.79 mL/kg/min) were assigned to the CON group. The uneven number of participants in the groups is due to the failure to follow up by two participants from the control group.

PEMF group used BEMER therapy 12 times across a 6-day period, before and after each training session. PEMF protocol included 8 min of laying on the BEMER mat, which transmitted a low-frequency electromagnetic field of flux density 35–50 µTesla (highest level) in a biorhythmic format.

### 2.5. Statistical Analyses

All results are expressed as mean ± standard deviation (SD). Independent t-test analysis was used to check for differences between groups at pretesting to confirm balanced groups after dropouts. Repeated measures mixed-model analysis was used to measure the difference between groups from pre- to post-testing using groups and time as a fixed factor and participants as a random factor and Tukey post hoc analysis. Mean differences are presented along with 95% confidence intervals (CI). Additionally, effect sizes were assessed to measure the magnitude of change within each group to measure the change from the elevated training camp, and between groups to measure the difference in change between PEMF and CON. All analyses were performed using SAS (Version 9.4, SAS Institute, Cary, NC, USA) and Prism (Version 8.4.0, GraphPad, San Diego, CA, USA).

## 3. Results

### Outcomes

Despite dropouts, both groups were similar at baseline for AbsVO_2Peak_ (*p* = 0.2497), RelVO_2Peak_ (*p* = 0.5685), VE (*p* = 0.1098), VT_Abs_ (*p* = 0.0965), VT_Rel_ (*p* = 0.3674), HR_Max_ (*p* = 0.3840), HR_VT_ (*p* = 0.2718), RR_Max_ (*p* = 0.3296), or RR_VT_ (*p* = 0.3577).There were no significant changes in AbsVO_2Peak_ (*p* = 0.1727) or RelVO_2Peak_ (*p* = 0.1149) for either the PEMF group or the CON group after the training camp;There was no significant difference for VE pre- to post-testing for any groups (*p* = 0.9305).There was a significant effect of time for both VT_Abs_ (*p* = 0.009) and VT_Rel_ (*p* ≤ 0.001). For PEMF, VT_Abs_ changed significantly from pretesting to post-testing (*p* = 0.001), but CON showed a nonsignificant trend towards difference (*p* = 0.061). Furthermore, VT_Rel_ was significantly different between pre- and post-tests for PEMF (*p* ≤ 0.001), and a nonsignificant difference was observed for CON (*p* = 0.098).HR_Max_ was significantly different from pre- to post-testing for both PEMF (*p* = 0.0212) and CON (*p* = 0.0251).Additionally, there was a significant time effect for HR_VT_ (*p* = 0.0326). However, the post hoc test showed that while PEMF had a significant difference (*p* < 0.0422), CON did not (*p* = 0.9477).There was no significant difference for RR_Max_ pre- to post-testing for any groups (*p* = 0.2557).There was a significant time effect for RR_VT_ (*p* = 0.0005).Table 2 and Figure 1 show the results for groups, while Figure 2 shows individual responses.

## 4. Discussion

The results of this pilot study revealed that daily use of BEMER-PEMF technology by athletes during a 6-day elevated preseason training camp did not induce significant changes in aerobic performance parameters, with the exception of VT_Rel_.

Considering that the only parameter that was significantly different between groups was VT_Rel_, the rest of the discussion will focus on this parameter. Based on our findings, the improvement was likely due to enhanced angiogenesis in the PEMF group. The majority of evidence supporting a link between PEMF and promotion of angiogenesis is from clinical studies [30,31,32,33], rather than an athletic setting. Nevertheless, because athletic performance, specifically high-intensity running, can cause hypoxic conditions within exercising skeletal muscle [34], translation of clinical findings to an athletic scenario may be viable. In particular, PEMF-induced angiogenesis has been observed in diseases where ischemia and hypoxia are major problems, such as myocardial infarction [30,35] peripheral arterial disease [31], and diabetes [36]. In these diseases, ischemia and subsequent tissue hypoxia are contributing factors to both the underlying pathogenesis and the symptomatic manifestations of the disease. Thus, approaches that attenuate ischemia via increased angiogenesis may prove therapeutic.

At a cellular level, multiple signaling pathways have been proposed as possible mediators of PEMF-induced angiogenesis [37]. Specifically, Pan et al. [31] demonstrated improved perfusion and neovascularization via PEMF in ischemic rat hind limbs, as demonstrated via laser Doppler perfusion imaging, by promoting signaling molecules such as FGF-2 and FGFR1. Additionally, others have demonstrated upregulation in Vascular endothelial growth factor (VEGF) via PEMF in human endothelial cells [32], cardiomyocytes [35], and mouse embryonic stem cells [38]. Finally, cell membrane adenosine receptors have been proposed as a biological pathway for the anti-inflammatory effects of PEMF [39]. Given the promotion of angiogenesis via PEMF in disease-related ischemia, there may be merit in the application of PEMF to enhance blood flow and perfusion to exercising muscle, as supported by our finding of improved VT via PEMF.

However, it is always hard to extrapolate data from animal models into human models, especially in the context of athletes; therefore, our speculation of angiogenesis needs to be confirmed in future studies. Another important point to note is that while Pan et al. [31] found a minor difference (not significant) after 7 days of using PEMF, there was a significant difference only after 14 days of usage in the rat model.

### Limitations

The lack of statistical significance seen in many dependent variables may be explained by insufficient sample size, due to the limited size of the running team and the additional runner dropouts. Future studies looking at the effects of PEMF application should therefore focus on larger sample sizes to confirm whether PEMF has an effect during elevated preseason training camp. In addition to assessing aerobic performance parameters, measuring hormonal/blood markers of physiological stress could provide further insight into the possible effects of PEMF on aerobic performance.

## 5. Conclusions

This study was the first of its kind to study PEMF technology in combination with preseason intensified training. Results indicate some evidence for the use of PEMF therapy during short-term training camps to improve VT. Further research is required to elucidate the physiological mechanisms of the PEMF-induced improvement in VT.

## Figures and Tables

**Figure 1 ijerph-18-07691-f001:**
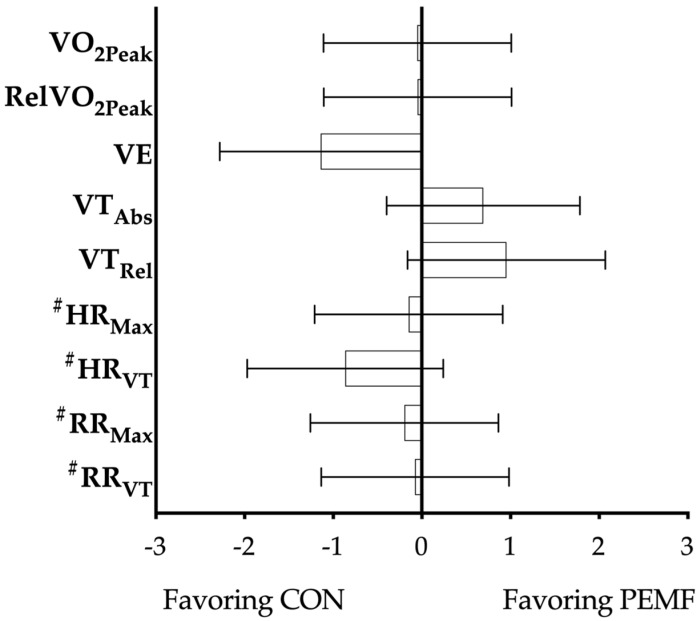
Effect sizes of differences between groups after 6-day elevated training camp, showing variables favoring either PEMF or CON group. Abbreviations: CON: control; PEMF: pulsed electromagnetic field; AbsVO_2Peak_: absolute VO_2Peak_; RelVO_2Peak_: relative VO_2Peak_; VE: ventilation; VT_Abs_: absolute ventilatory threshold; VT_Rel_: relative ventilatory threshold; HR_Max_: maximum heart rate; HR_VT_: heart rate @ ventilatory threshold; RR_Max_: maximum respiration rate; RR_VT_: respiration rate @ ventilatory threshold. ^#^ indicates that a negative change was favorable.

**Figure 2 ijerph-18-07691-f002:**
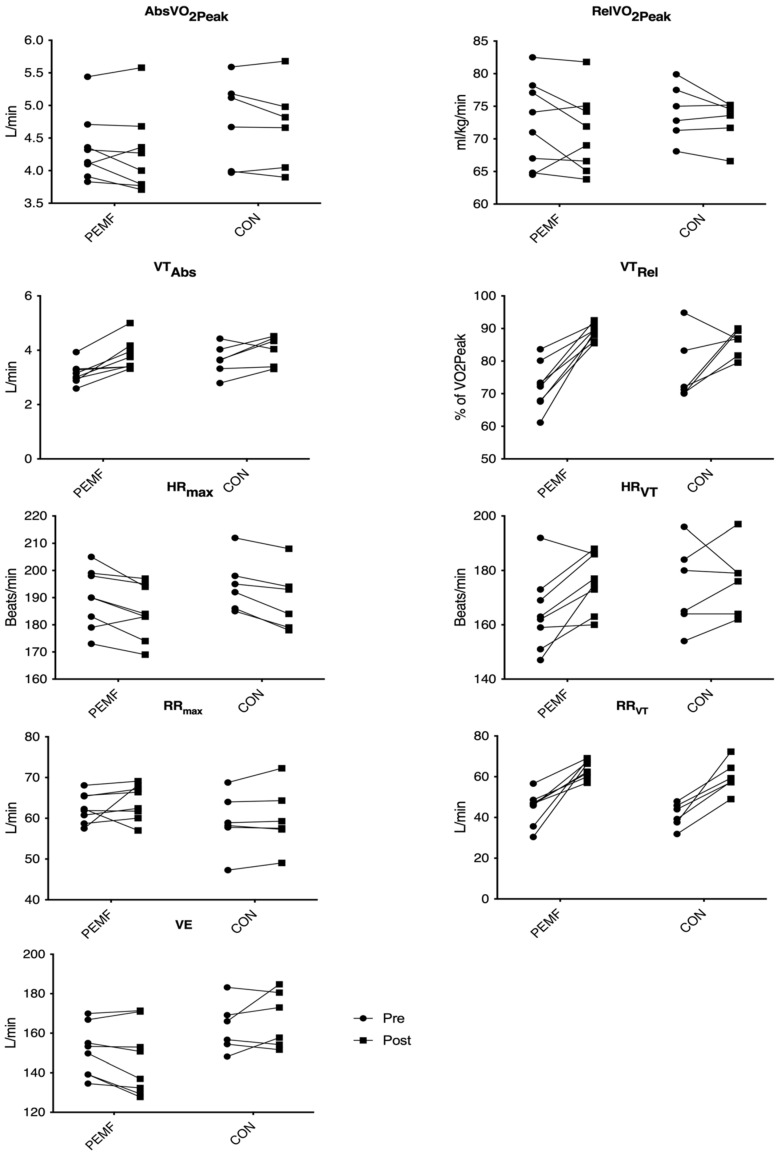
Individual responses of physiological variables at peak exercise and ventilatory threshold of the participants. Abbreviations: CON: control; PEMF: pulsed electromagnetic field; AbsVO_2Peak_: absolute VO_2Peak_; RelVO_2Peak_: relative VO_2Peak_; VE: Ventilation; VT_Abs_: absolute ventilatory threshold; VT_Rel_: relative ventilatory threshold; HR_Max_: maximum heart rate; HR_VT_: heart rate @ ventilatory threshold; RR_Max_: maximum respiration rate; RR_VT_: respiration rate @ ventilatory threshold.

**Table 1 ijerph-18-07691-t001:** Daily training regime.

Day	Distance	Time	Altitude_Min_	Altitude_Max_	Altitude_Avg_	Speed_Avg_
	m	mins	m	m	m	m/s
Day 1	16,093.44	74	1007.36	1205.79	1106.58	3.62
Day 2	16,093.44	71	753.77	809.85	781.81	3.78
Day 3	19,312.13	76.5	892.30	902.82	897.5598	4.21
Day 4	19,312.13	87	667.82	822.05	744.93	3.70
Day 5	16,415.31	77	1061.31	1388.36	1224.84	3.55
Day 6	11,265.41	66	904.95	1157.33	1031.14	2.84

**Table 2 ijerph-18-07691-t002:** Changes for cardiopulmonary variables between intervention and control groups.

		PEMF				CON			
		Pre	Post	MD	95% CI	Pre	Post	MD	95% CI
AbsVO_2Peak_	L/min	4.35 ± 0.52	4.27 ± 0.63	−0.08 ± 0.07	−0.23 to 0.07	4.75 ± 0.66	4.68 ± 0.65	−0.07 ± 0.08	−0.24 to 0.10
RelVO_2Peak_	mL/kg/min	72.40 ± 6.67	70.94 ± 6.03	−1.46 ± 1.06	−3.77 to 0.84	74.10 ± 4.28	72.82 ± 3.32	−1.28 ± 1.22	−3.94 to 1.38
VE	L/min	150.98 ± 13.00	146.58 ± 17.78	−4.4 ± 2.62	−10.11 to 1.31	162.96 ± 12.58	167.00 ± 14.30	4.04 ± 3.03	−2.55 to 10.64
VT_Abs_	L/min	3.14 ± 0.39	3.80 ± 0.57 *	0.66 ± 0.15	0.33 to 1.00	3.64 ± 0.57	4.01 ± 0.54	0.37 ± 0.18	−0.02 to 0.75
VT_Rel_	% of RelVO_2Peak_	72.38 ± 7.14	88.95 ± 2.41 *	16.58 ± 2.95	10.16 to 22.99	76.97 ± 10.07	85.72 ± 4.19 ^	8.75 ± 3.40	1.34 to 16.16
HR_Max_	Beats/min	189.63 ± 10.88	184.88 ± 10.08 *	−4.75 ± 1.37	−7.74 to −1.76	194.67 ± 9.87	189.33 ± 11.38 *	−5.33 ± 1.58	−8.79 to −1.88
HR_VT_	Beats/min	164.50 ± 14.02	176.00 ± 10.56 *	11.50 ± 3.75	3.33 to 19.67	173.83 ± 15.50	176.17 ± 12.64	2.33 ± 4.33	−7.10 to 11.77
RR_Max_	Breaths/min	62.56 ± 3.62	64.02 ± 4.33	1.46 ± 1.23	−1.22 to 4.14	59.18 ± 7.22	59.97 ± 7.79	0.78 ± 1.42	−2.31 to 3.88
RR_VT_	Breaths/min	44.69 ± 8.10	54.04 ± 4.29 *	9.35 ± 2.53	0.01 to 3.84	41.09 ± 5.97	49.89 ± 8.66 *	8.80 ± 2.92	2.44 to 15.16

Note: Values are mean ± standard deviation. Abbreviations: PEMF: pulsed electromagnetic field; CON: control; MD: Mean difference; CI: confidence interval; AbsVO_2Peak_: absolute VO_2Peak_; RelVO_2Peak_: relative VO_2Peak_; VE: ventilation; VT_Abs_: absolute ventilatory threshold; VT_Rel_: relative ventilatory threshold; HR_Max_: maximum heart rate; HR_VT_: heart rate @ ventilatory threshold; RR_Max_: maximum respiration rate; RR_VT_: respiration rate @ ventilatory threshold. * indicates significantly different, compared to pre (*p* < 0.05). ^ indicates trending toward significant, compared to pre (0.05 < *p* < 0.10).

## Data Availability

The data is available by request from the corresponding author.

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
