# Peer review of "Effects of Acute Low-Frequency Pulsed Electromagnetic Field Therapy on Aerobic Performance during a Preseason Training Camp: A Pilot Study"

_ijerph, 2021, doi:10.3390/ijerph18147691_

Round 1

Reviewer 1 Report

The authors investigate the effect of Bio-Electro-Magnetic-Energy-Regulation on the endurance athletes during a pre-season training.

The study is interesting. The stated work limit is a smaller number of study participants.

There are shortcomings in the literature list (for example lines: 283, 332, 333).

The training content in the reporting period was the same every day?

How did running training change during the intervention, as opposed to training prior to this period? Has there been a change in the intensity or volume of training in the reporting period? Can this explain the reduction in VO2max during the intervention period?

As the intervention group experienced a decrease in VE and a significant increase in VT, a decrease in respiratory rate had to occur. I believe it would be worth publishing this parameter. This would suggest to efficiency increasing of the breathing process.

Author Response

Kindly thank you for your remarks.

The study is interesting. The stated work limit is a smaller number of study participants.

There are shortcomings in the literature list (for example lines: 283, 332, 333).

Answer: The literature style was revised accordingly.

The training content in the reporting period was the same every day?

Answer: We have added training content of each day of training in table 1.

How did running training change during the intervention, as opposed to training prior to this period? Has there been a change in the intensity or volume of training in the reporting period? Can this explain the reduction in VO2max during the intervention period?

Answer: Table 1 and lines 157-158 are answering these questions. Reduction of VO2max is not significant (Table 2).

As the intervention group experienced a decrease in VE and a significant increase in VT, a decrease in respiratory rate had to occur. I believe it would be worth publishing this parameter. This would suggest to efficiency increasing of the breathing process.

Answer: We added respiratory rate data (table 2, figures 1 and 2).  Explained also in chapter 3.1

Reviewer 2 Report

With the corrections, the paper has improved.

Clarification of “A Pilot Study” is necessary.

Are they fourteen (in abstract) o sixteen (in participants) athletes? The loss of two athletes in the study is not clarified.

But it has important limitations to accept the paper in this journal.

In addition to those commented by the authors, one of the most important limitations is the assessment of altitude.

The authors say that the altitude between 881.83 (± 20 135.98 m) to 1027.0 (± 223.44 m) above sea-level is high. This altitude is low to be able to obtain significant results (2000- 25000 m being considered optimal in endurance athletes).

Altitude is not a relevant data in this study.

Author Response

Kindly thank you for your remarks.

Clarification of “A Pilot Study” is necessary.

Answer: Revised and explained – lines 88-91.

Are they fourteen (in abstract) o sixteen (in participants) athletes? The loss of two athletes in the study is not clarified.

Answer: Revised. Initially 16 started but two failed to follow up. Results are presented for 14 participants.

In addition to those commented by the authors, one of the most important limitations is the assessment of altitude.

The authors say that the altitude between 881.83 (± 20 135.98 m) to 1027.0 (± 223.44 m) above sea-level is high. This altitude is low to be able to obtain significant results (2000- 2500 m being considered optimal in endurance athletes).

Answer: We agree with your comments. We deleted from text low altitude as much as we can and replaced with elevation. We believe it is necessity to include conditions and workload of training therefore we added Table 1.

Altitude is not a relevant data in this study.

Answer: Our findings are not related with altitude however we more clearly explained conditions of the training camp.

Revised paper with revisions in color is attached. 

Round 2

Reviewer 2 Report

The authors have improved the manuscript and added relevant information for its understanding.

I think it can be published in the current form.

This manuscript is a resubmission of an earlier submission. The following is a list of the peer review reports and author responses from that submission.

Round 1

Reviewer 1 Report

The authors investigate the effect of Bio-Electro-Magnetic-Energy-Regulation on the endurance athletes during a pre-season training.

The study is interesting. The stated work limit is a smaller number of study participants.

Line 76, 77, 207, 219: Verify citations in the text, commonly is used, for example: Grote et al. [23] who showed ...

Line 136: It would be useful to indicate whether the control group had received the same training load, it would also be appropriate to describe the training load participants had received.

I recommend explaining the observed changes in each parameter for both groups, especially the control group, what explains these changes?

Line 176: Mark 3.1 repeated

Line 182: Missing table name

Line 184: You are mentioning effect size in abbreviations but you did not involve it in the table. 

Line 233: It is appropriate to better describe and present the findings of the research.

It would be useful to explain the supposed reasons for the increase in VT, what is causing it and the practical relevance of this results.

Line 259: Please try to provide more updated studies.

Reviewer 2 Report

While I commend the authors for conducting this study, the interpretation of the data is a concern. Although it is a preliminary study that begins a new field of research, I consider that there are several points where the study should improve.

The comments for the study are as follows:

To justification about the use of BEMER- PEMF technology, explanations from manufacturers are used that have not yet been proven (lines 46-49), which seems like a marketing point of view versus science. After have read those lines in detail, they describe the effects demonstrated in studies.

The statement made between lines 60-61 “The positive effects seen from PEMF therapy have mostly been seen in extreme situations, while effects for healthy adults have been inconclusive”, is it used to relate the possible response of PEMF technology in extreme situations such as the exposure of athletes to high altitude training? (detailed on subsequent lines). The statement of lines 63-65 “The onset of such a training regimen, in addition to an elevated altitude, following a period of sedentary behavior at sea-level, represents a significant physical challenge to athletes” relates to the training regimen with elevated altitude. This is a true statement, but it does not happen in their study since athletes train at very low altitudes, which may not cause any changes or important modifications at a physiological level.

The altitudes used to find answers at the performance level, range from 1400- 2500 m, 2000-2500 m being considered optimal in endurance athletes (Chapman et al., 2014).

I recommend reading (Mujika et al., 2019), among other papers, to deepen on altitude training in endurance athletes.

Why is it specified in the study that it is training at low altitude if that is not going to involve important modifications in the runners?

In addition, the application period of 6 days is reduced to be able to obtain reliable conclusions (improvement of VT). And the sample size is small.

The information provided by the evaluated parameters is limited. Moreover, as the authors describe in the limitations, to assess hormonal and blood parameters can demonstrate the possible recovery effect of BEMER- PEMF technology on aerobic performance.

From my view, the study has an interesting part, which is the use of BEMER technology to improve performance in endurance athletes, however, it have several limitations to be able to make real claims.

Chapman, R. F., Karlsen, T., Resaland, G. K., Ge, R.-L., Harber, M. P., Witkowski, S., Stray-Gundersen, J., & Levine, B. D. (2014). Defining the “dose” of altitude training: how high to live for optimal sea level  performance enhancement. Journal of Applied Physiology (Bethesda, Md. : 1985), 116(6), 595–603. https://doi.org/10.1152/japplphysiol.00634.2013

Mujika, I., Sharma, A. P., & Stellingwerff, T. (2019). Contemporary Periodization of Altitude Training for Elite Endurance Athletes: A  Narrative Review. Sports Medicine (Auckland, N.Z.), 49(11), 1651–1669. https://doi.org/10.1007/s40279-019-01165-y